# Analytical Solutions for Simple Turbulent Shear Flows on a Basis of a Generalized Newton’s Law

**DOI:** 10.3390/polym14163308

**Published:** 2022-08-14

**Authors:** Dmitry Nikushchenko, Valery Pavlovsky, Elena Nikushchenko

**Affiliations:** 1Institute of Hydrodynamics and Control Processes, Saint-Petersburg State Marine Technical University, Lotsmanskaya 3, 190121 Saint-Petersburg, Russia; 2Research Department, Saint-Petersburg State Marine Technical University, Lotsmanskaya 3, 190121 Saint-Petersburg, Russia

**Keywords:** rheology, non-Newtonian fluids, power relation, resistance, Newton’s formula, Blasius formula, Toms effect, laminar flow, turbulent flow, simple share flow

## Abstract

In the presented article a generalization of Newton’s formula for the shear stress in a fluid is carried out by giving it a power-law form. After the introduction of the corresponding strain rate tensor, a generalization is made to the spatial case of flow and the rheological relation is presented in tensor form. Depending on the power value in this rheological ratio, one can come either to a description of a laminar flow regime (in the form of Navier–Stokes equations), or to a description of the flow in turbulent regime. In the latter case, a set of differential equations with the no-slip boundary condition is specified, which is significantly different from that for the laminar flow regime, but which also allows one to obtain analytical solutions for simple shear flows and obtain the Blasius resistance law for the flow in a pipe. Therefore, the considered approach to solving problems of turbulent flows compares favorably with modern differential turbulence models. Solutions are given for simple shear flows of a fluid, when there is only one longitudinal component of the velocity, which depends on the transversal coordinate only. These solutions in terms of velocity profiles and resistance coefficients are in satisfactory agreement with the experimental data.

## 1. Introduction

The formation of hydrodynamics as a science is largely associated with research of fluid flow in pipes [1,2]. A round pipe is the most common hydrodynamic object used for theoretical solutions, as a huge amount of experimental data has been accumulated for the flow in the round pipe, which makes it possible to evaluate the quality of one or another obtained formulas for resistance coefficients and velocity profiles. At the same time, unlike the laminar flow, there are no exact analytical solutions for the turbulent flow regime and all formulas and relations are either empirical or semi-empirical in nature, which is based on the concept of turbulent viscosity. With the help of this concept, L. Prandtl [2] developed the theory of mixing length, the first phenomenological theory of turbulence, which has been applied in the calculations of turbulent flow in pipes. The theory made it possible to obtain methods for solving many applied problems, which are important for engineering applications. Subsequently, semi-empirical theories of turbulence, based on the concept of turbulent viscosity, were developed intensively and now they have acquired the form of differential models of turbulence ($“k-\epsilon^{”}$, $“k-\omega^{”}$, etc.) [3]. For the numerical solution of such problems, various methods could be used, including grid methods, which have high computational complexity. At the same time, methods of parallelizing grid methods on multiprocessor computing systems with shared and distributed memory are used. The main methods for solving problems of near-wall flows can be found, for example, in the articles [4,5].

The empirical relation used in the modern theory of turbulence is a universal velocity profile suitable for describing any near-wall fluid flows. It consists of several regions, and each region is described by a corresponding formula. The central region in it is the logarithmic one, the formula for which is given by Prandtl’s theory. This formula cannot be extended to the flow zone near the wall, where the no-slip condition takes place. For this reason, L. Prandtl and then T. Karman drew attention to the possibility of describing the velocity profile using the following power-law formula: the dependency of velocity on the distance to the wall in the 1/7 power [1,6]. That law was called «1/7 law», and it provides the velocity distribution that ensures that the no-slip condition is satisfied and leads to the power law of resistance in the Blasius formula form. The disadvantage of this law, like all power-law velocity profiles, is that it leads to an infinite derivative of the velocity on the wall and does not give a zero value of the derivative of the velocity on the pipe axis. In general, this power-law formula for the distribution of averaged velocities in a pipe is not worse than the logarithmic one, but both are unsuitable for describing the behavior of the flow close to the pipe axis [1].

Power-law formulas have become widespread in hydrodynamics both in the study of turbulent fluid flows and in the rheological relations for non-Newtonian fluids [7]. For non-Newtonian fluids, when describing near-wall flows, the shear stress τ is presented [8,9,10,11] as power-law dependency on the velocity gradient $\dot{\gamma }=\frac{du}{dy}$, where *u* is the longitudinal velocity of a fluid particle and *y* is the transversal coordinate:(1)$$\tau =k\dot{\gamma }{\;}^{n}.$$

Here *n* is the power, a non-dimensional quantity, *k*, [$Pa\cdot sec^{n}$] is the proportion ratio. Quantities *n* and *k* are experimentally determined for each type of non-Newtonian fluid. A fairly large number of papers [8,11] are currently devoted to the generalization of Formula (Equation 1) to the three-dimensional case of flow in tensor form.

In the theory of turbulence, the Blasius power-law formula is widely used for the resistance coefficient of a fluid flow with kinematic viscosity ν in a straight round pipe of radius *R* [1,8]:(2)$$\lambda =\frac{0.3164}{Re^{\frac{1}{4}}},$$
where $Re=\frac{2Ru_{av}}{\nu }$ is the Reynolds number, calculated by the average flow velocity $u_{av}$. From considerations of dimensionality, T. Karman showed that this power-law of resistance corresponds to a power-law velocity profile:(3)$$\frac{u}{u_{max}}={\left(\frac{y}{R}\right)}^{\frac{1}{7}},$$
where *y* is the distance between the pipe axis and the wall. Blasius’ formula is a special case of the general power-law resistance law: $\lambda =\frac{Const}{Re^{m}}$, which corresponds to the power law of velocity $\frac{u}{u_{max}}={\left(\frac{y}{R}\right)}^{n}$. There is a relation between *m* and *n* parameters [1], which ultimately leads to the law of resistance $\lambda =\frac{Const}{Re^{2n/(n+1)}}$. Now if we assume $n=\frac{1}{7}$ the Blasius’ formula $\lambda =\frac{Const}{Re^{0.25}}$ will be obtained.

Interest in the flow in round pipes in the turbulent flow regime is still shown at the present time. This is due to the fact that for engineering applications, the most important task is to assess the head losses to overcome the hydraulic resistances arising from the movement of fluids in the pipelines. Accurate accounting of these losses largely determines the reliability of technical calculations, the degree of perfection and economic feasibility of engineering decisions taken during their design. Therefore, a search for more and more perfect methods for calculating flows in pipes is still relevant. At the same time, along with the algebraic rheological relations using the L. Prandtl foundation, modern differential models of turbulence are also used [12,13] and even power-law formulas for velocity profiles [14,15].

## 2. Generalization of Newton’s Formula

Newton’s law for shear stress in flow around a flat surface is as follows:(4)$$\tau =\rho \nu \frac{du}{dy},$$
where ρ is the density, ν is the kinematic viscosity; the Formula (Equation 4) can be generalized as follows [16]:(5)$$\tau =\rho \chi_{n}\;{\left(\nu \frac{du^{2n-1}}{dy}\right)}^{\frac{1}{n}}.$$

In this expression, the power *n* may take n≥1 values, $\chi_{n}$ is a non-dimensional coefficient, depending on this power value. For n=1 when $\chi_{n}=1$, the Formula (Equation 5) leads to Newton’s rheological ratio and, as a result, to Poiseuille formula for laminar flow in the pipe. For n=4 and $\chi_{n}=0.019746$, this formula leads to the rheological relation for turbulent fluid flow in the pipe and then to the Blasius’ formula for resistance coefficient. Formula (Equation 5) can also be represented in the following form, convenient for further generalization:(6)$$\tau =\rho \chi_{n}\;{\left(2n-1\right)}^{1/n}{\left(u^{2}\right)}^{\frac{n-1}{n}}{\left(\nu \frac{du}{dy}\right)}^{\frac{1}{n}}.$$

A generalization of the Formula (Equation 5) to the spatial case of flow makes it possible to acquire the corresponding rheological relation in tensor form [17]. As a result, the system of differential equations arises, similar to the system of Navier–Stokes equations, which makes it possible to solve boundary value problems in fluid mechanics. The obtained system of equations for arbitrary values of the power can also be used to describe the behavior of fluids under various flow regimes of both Newtonian and non-Newtonian fluids. For arbitrary values of the power *n* and the corresponding values of the $\chi_{n}$ coefficient, this system has the following form in the Cartesian rectangular coordinate system, disregarding the mass forces:(7)$$\rho \frac{dV_{i}}{dt}=-\frac{\partial p}{\partial x_{i}}+B\frac{\partial }{\partial x_{k}}\left[{\left(\frac{V_{j}V_{j}}{\sqrt{Y_{2}}}\right)}^{\frac{n-1}{n}}\left(\frac{\partial V_{i}}{\partial x_{k}}+\frac{\partial V_{k}}{\partial x_{i}}\right)\right],$$
where the following notations are introduced for brevity:(8)$$B=\rho \chi_{n}{\left(2n-1\right)}^{\frac{1}{n}}\nu^{\frac{1}{n}},$$
and $Y_{2}$ is the second invariant of the strain rate tensor $\underline{S}=\frac{1}{2}\left(\nabla \vec{V}+\nabla {\vec{V}}^{T}\right)$, $\vec{V}$ is the velocity vector, *∇* is the Hamilton operator, $“T^{”}$ is the transportation symbol [17]:(9)$$Y_{2}=\frac{1}{2}\;\underline{S}:\underline{S}=\frac{1}{2}S_{ij}\;S_{ji},$$
or in expanded form:$$\begin{array}{ccc} Y_{2} & = & \frac{1}{2}\left[{\left(\frac{\partial V_{x}}{\partial x}\right)}^{2}+{\left(\frac{\partial V_{y}}{\partial y}\right)}^{2}+{\left(\frac{\partial V_{z}}{\partial z}\right)}^{2}\right]+\\ & + & {\left(\frac{\partial V_{x}}{\partial y}+\frac{\partial V_{y}}{\partial x}\right)}^{2}+{\left(\frac{\partial V_{y}}{\partial z}+\frac{\partial V_{z}}{\partial y}\right)}^{2}+{\left(\frac{\partial V_{x}}{\partial z}+\frac{\partial V_{z}}{\partial x}\right)}^{2}\;. \end{array}$$

For each power *n* value, the non-dimensional parameter $\chi_{n}$ is determined by experience.

For n=1 and $\chi_{n}=1$, there is a common Navier–Stokes equation describing the laminar flow regime:(10)$$\rho \frac{dV_{i}}{dt}=-\frac{\partial p}{\partial x_{i}}+\rho \nu \left(\frac{\partial^{2}V_{i}}{\partial x_{k}\partial x_{k}}+\frac{\partial^{2}V_{k}}{\partial x_{k}\partial x_{i}}\right).$$

A particular case of Equation (Equation 7) for n=4 and $\chi_{n}=0.019746$ would be the differential equation describing turbulent flows for the “Blasius” range of Reynolds numbers (for a flow in a pipe, which leads to the Blasius’ formula when $10^{4}<Re<10^{6}$):(11)$$\rho \frac{dV_{i}}{dt}=-\frac{\partial p}{\partial x_{i}}+\rho \chi_{n}{\left(7\nu \right)}^{\frac{1}{4}}\frac{\partial }{\partial x_{k}}\left[{\left(\frac{V_{j}V_{j}}{\sqrt{Y_{2}}}\right)}^{\frac{3}{4}}\left(\frac{\partial V_{i}}{\partial x_{k}}+\frac{\partial V_{k}}{\partial x_{i}}\right)\right].$$

In projections on the axises of the Cartesian rectangular coordinate system, we have a system of three partial differential equations:(12)$$\begin{array}{cc} \; & \rho \frac{dV_{x}}{dt}=-\frac{\partial p}{\partial x}+C\left[\frac{\partial }{\partial x}\Phi \left(2\frac{\partial V_{x}}{\partial x}\right)+\frac{\partial }{\partial y}\Phi \left(\frac{\partial V_{x}}{\partial y}+\frac{\partial V_{y}}{\partial x}\right)+\frac{\partial }{\partial z}\Phi \left(\frac{\partial V_{x}}{\partial z}+\frac{\partial V_{z}}{\partial x}\right)\right],\\ \; & \rho \frac{dV_{y}}{dt}=-\frac{\partial p}{\partial y}+C\left[\frac{\partial }{\partial x}\Phi \left(\frac{\partial V_{y}}{\partial x}+\frac{\partial V_{x}}{\partial y}\right)+\frac{\partial }{\partial y}\Phi \left(2\frac{\partial V_{y}}{\partial y}\right)+\frac{\partial }{\partial z}\Phi \left(\frac{\partial V_{y}}{\partial z}+\frac{\partial V_{z}}{\partial y}\right)\right],\\ \; & \rho \frac{dV_{z}}{dt}=-\frac{\partial p}{\partial z}+C\left[\frac{\partial }{\partial x}\Phi \left(\frac{\partial V_{z}}{\partial x}+\frac{\partial V_{x}}{\partial z}\right)+\frac{\partial }{\partial y}\Phi \left(\frac{\partial V_{z}}{\partial y}+\frac{\partial V_{y}}{\partial z}\right)+\frac{\partial }{\partial z}\Phi \left(2\frac{\partial V_{z}}{\partial z}\right)\right]. \end{array}$$

For the notation brevity, it is denoted that:(13)$$C=\rho \chi_{n}{\left(7\nu \right)}^{\frac{1}{4}},\;\;\;\Phi ={\left(\frac{V_{x}^{2}+V_{y}^{2}+V_{z}^{2}}{\sqrt{Y_{2}}}\right)}^{\frac{3}{4}}.$$

To close the set of Equation (Equation 12) in the case of an incompressible fluid flow, it is also necessary to add continuity equation $\nabla \cdot \vec{V}=0$, which in coordinate form is as follows:(14)$$\frac{\partial V_{x}}{\partial x}+\frac{\partial V_{y}}{\partial y}+\frac{\partial V_{z}}{\partial z}=0.$$

The boundary condition for this system of equations is the no-slip condition. It is necessary to note that, as with all other power rheological relations, the proposed generalization of Newton’s formula is not invariant with respect to the Galilean transformation. Therefore, the area of possible applications of the set of Equations (Equation 12)–(Equation 14) is limited to cases of flow over stationary walls.

Problems of turbulent fluid flows around bodies of random geometry can be solved on the basis of Equation (Equation 12). The calculation results will be valid and agree with the experimental data for a relatively narrow “Blasius” range of Reynolds numbers, which, for example, is $10^{4}<Re<10^{6}$ for the flow in a pipe and $10^{5}<Re<10^{8}$ for flow over a flat plate.

For steady-state simple shear flows, there is one longitudinal velocity component $V_{x}$, which depends on one transversal coordinate *y*. Denoting $V_{x}=u$ from the first equation of (Equation 12) equation and considering that $Y_{2}={\left(\frac{du}{dy}\right)}^{2}$, $\sqrt{Y_{2}}=\left|\frac{du}{dy}\right|$:(15)$$0=-\frac{dp}{dx}+\rho \chi_{n}{\left[\nu \left(2n-1\right)\right]}^{1/n}\frac{d}{dy}\left[{\left(\frac{u^{2}}{\left|\frac{du}{dy}\right|}\right)}^{\frac{n-1}{n}}\frac{du}{dy}\right].$$

When $\frac{du}{dy}\geq 0$ and the modulus of the velocity derivative is positive, the expression in square brackets in (Equation 15) for the near-wall simple shear flow can be transformed as follows:$${\left(\frac{u^{2}}{\left|\frac{du}{dy}\right|}\right)}^{\frac{n-1}{n}}\frac{du}{dy}={\left(u^{2(n-1)}\frac{du}{dy}\right)}^{\frac{1}{n}}={\left(\frac{1}{2n-1}\frac{du^{2n-1}}{dy}.\right)}^{\frac{1}{n}}.$$

As a result, for such near-wall simple shear flow, the equation of motion (Equation 15) takes the form:(16)$$-\frac{dp}{dx}+\rho \chi_{n}\frac{d}{dy}{\left[\nu \frac{du^{2n-1}}{dy}\right]}^{\frac{1}{n}}=0.$$

In the case of negative $\frac{du}{dy}$ after expanding the modulus in the Equation (Equation 15), further similar transformations should be performed.

## 3. Steady State Flow in a Circular Cylindrical Pipe at Arbitrary Value of the *n* Power

A steady flow in a circular cylindrical pipe of radius *R*, with transversal coordinate *y*, which is measured from the wall (0≤y≤R) was studied in the article [16]. Here we briefly describe the main results obtained. Non-dimensional coordinate and velocity are introduced:(17)$$\left\{\begin{array}{c} \eta =\frac{y}{R},\;\;0\leq \eta \leq 1,\\ V=\frac{u}{V_{*}}, \end{array}\right.$$
where $V_{*}$ is the friction velocity, expressed in terms of shear stress $\tau_{w}$ on the wall, obtained from the equation of fluid motion in stresses [18]:(18)$$\left|\tau_{w}\right|=\frac{d}{4}\frac{\Delta p}{l},\;\;\;V_{*}=\sqrt{\frac{\left|\tau_{w}\right|}{\rho }}.$$

In these expressions, d=2R is the pipe diameter, Δp is the longitudinal pressure drop along the pipe of *l* length. Then, considering notations (Equation 17), expression (Equation 5) can be represented as follows:(19)$$\tau =\rho \chi_{n}{\left(\nu \;\frac{{V_{*}}^{2n-1}}{R}\frac{dV^{2n-1}}{d\eta }\right)}^{\frac{1}{n}}=\rho \chi_{n}V_{*}^{2}\;{\left(\;\frac{\nu }{V_{*}R}\frac{dV^{2n-1}}{d\eta }\right)}^{\frac{1}{n}}.$$

This non-negative expression can also be associated with the non-negative value of the shear stress $\tau =\tau_{w}\left(1-\frac{y}{R}\right)$ obtained from equation of motion of the continuous medium in stresses, i.e., the $\rho V_{*}^{2}\left(1-\eta \right)$ parameter:(20)$$\rho \chi_{n}V_{*}^{2}\;{\left(\;\frac{\nu }{V_{*}R}\frac{dV^{2n-1}}{d\eta }\right)}^{\frac{1}{n}}=\rho V_{*}^{2}\left(1-\eta \right).$$

Then:(21)$$\frac{dV^{\left(2n-1\right)}}{d\eta }=\frac{Re_{*}}{\chi_{n}^{n}}{\left(1-\eta \right)}^{n},$$
where $Re_{*}=\frac{V_{*}R}{\nu }$ is Reynolds number, computed by the friction velocity.

The boundary condition for this equation is the no-slip condition: η=0, V=0.

Integration of Equation (Equation 21) with this boundary condition leads to the following expression:$$V^{2n-1}=\frac{Re_{*}}{\chi_{n}^{n}\left(n+1\right)}\left[1-{\left(1-\eta \right)}^{n+1}\right],$$
from which non-dimensional velocity profile takes the form:(22)$$V={\left(\frac{Re_{*}}{\left(n+1\right)\chi_{n}^{n}}\right)}^{\frac{1}{2n-1}}{\left[1-{\left(1-\eta \right)}^{n+1}\right]}^{\frac{1}{2n-1}}.$$

The average non-dimensional velocity over the pipe cross section [8,19],
$$V_{av}=2\int_{0}^{1}V\left(\eta \right)\left(1-\eta \right)d\eta ,$$
leads to the following expression:(23)$$V_{av}=2{\left(\frac{Re_{*}}{\left(n+1\right)\chi_{n}^{n}}\right)}^{\frac{1}{2n-1}}Y\left(n\right),$$
where Y(n) has the following form for a flow in a circular cylindrical pipe:(24)$$Y\left(n\right)=\int_{0}^{1}{\left[1-{\left(1-\eta \right)}^{n+1}\right]}^{\frac{1}{2n-1}}\left(1-\eta \right)d\eta .$$

This parameter is expressed through hypergeometric functions, its values for different power values are shown in the Table 1. For other *n* values, Y(n) can be found in reference materials. It is useful to note that when n→∞, Y(n)→0.5.

True Reynolds number $Re=\frac{2RV_{av}}{\nu }$ computed by the non-dimensional average velocity $V_{av}$ can be calculated [18,19] in terms of $Re_{*}$ number using friction velocity:$$Re=2Re_{*}V_{av}.$$

Then, taking into account the Formula (Equation 23), we have:(25)$$Re=4Y\left(n\right){\left(\frac{1}{\left(n+1\right)\chi_{n}^{n}}\right)}^{\frac{2n}{2n-1}}Re_{*}^{\frac{2n}{2n-1}}.$$

Hence, we can express $Re_{*}$ inversely in terms of Re:(26)$$Re_{*}=\frac{{\left(\left(n+1\right)\chi_{n}^{n}\right)}^{\frac{1}{2n}}}{{\left(4Y(n)\right)}^{\frac{2n-1}{2n}}}\;Re^{\frac{2n-1}{2n}}.$$

The square of the non-dimensional average velocity is written as:(27)$$V_{av}^{2}=\frac{2^{\frac{2(n-1)}{n}}\;Y{\left(n\right)}^{\frac{2n-1}{n}}}{\chi_{n}{\left(n+1\right)}^{\frac{1}{n}}}\;Re^{\frac{1}{n}}.$$

Resistance coefficient $\lambda =\frac{8\left|\tau_{w}\right|}{\left(\rho u_{av}^{2}\right)}\;$ can be represented in terms of the square of the non-dimensional average velocity as $\lambda =\frac{8}{V_{av}^{2}}$, which, considering (Equation 23) expression, leads to a formula for this coefficient at an arbitrary *n* value:(28)$$\lambda =2^{\frac{n+2}{n}}\;\frac{{\left(n+1\right)}^{\frac{1}{n}}}{Y{\left(n\right)}^{\frac{2n-1}{n}}}\;\frac{\chi_{n}}{Re^{\frac{1}{n}}}.$$

Expressions (Equation 23)–(Equation 28) make it possible to describe the velocity field and resistance coefficient of a flow in the straight round pipe for any values of *n* power.

## 4. Special Cases for Different Values of the *n* Power

In special cases of n=1 and n=4, the obtained formulas lead to expressions for the laminar and turbulent flow regimes, respectively. Let us consider these cases in details.

When n=1 and $\chi_{n}=1$, formula (Equation 5) leads to Newton’s viscous friction formula:(29)$$\tau =\rho \nu \frac{du}{dy}.$$

In this case, the non-dimensional velocity profile according to (Equation 22), takes the form:(30)$$V=\frac{Re_{*}}{2}\left[1-{\left(1-\eta \right)}^{2}\right].$$

This formula corresponds to the Poiseuille profile, which can be obtained after the transition to cylindrical coordinates:$$\eta =\frac{y}{r}=\frac{R-r}{R}=1-\frac{r}{R},$$
$$\begin{array}{ccc} u & = & \frac{\frac{1}{2}RV_{*}}{\nu }\left(\frac{R^{2}-r^{2}}{R^{2}}\right)V_{*}=\frac{1}{2}\frac{R}{\nu }V_{*}^{2}\left(\frac{R^{2}-r^{2}}{R^{2}}\right)=\\ & = & \frac{1}{2}\frac{R}{\nu }\frac{1}{2\rho }\frac{\delta p}{l}R\left(\frac{R^{2}-r^{2}}{R^{2}}\right)=\frac{1}{4\mu }\left(-\frac{dp}{dz}\right)\left(R^{2}-r^{2}\right). \end{array}$$

Non-dimensional velocity averaged over the pipe cross section according to expression (Equation 27), considering that Y(1)=0.25, is:$$V_{av}=0.25Re,$$
and the relation between the Reynolds numbers according to (Equation 25) will be
$$Re=\frac{1}{2}Re_{*}^{2}.$$

The resistance coefficient according to expression (Equation 27) for n=1 and $\chi_{n}=1$ will be as follows:(31)$$\lambda =\frac{64}{Re},$$
as was expected [1,20,21]. Thus, for n=1 and $\chi_{n}=1$, rheological relation (Equation 5) describes the laminar flow regime.

For n=4 and $\chi_{n}=0.019746$, rheological relation (Equation 5) takes the form:(32)$$\tau =\rho \chi_{n}{\left(\nu \frac{du^{7}}{dy}\right)}^{\frac{1}{4}}.$$

Non-dimensional velocity profile according to (Equation 22) is as follows:(33)$$V={\left(\frac{Re_{*}}{5\chi_{n}^{4}}\right)}^{\frac{1}{7}}\;{\left[1-{\left(1-\eta \right)}^{5}\right]}^{\frac{1}{7}}.$$

Cross-section average velocity:(34)$$V_{av}=2{\left(\frac{Re_{*}}{5\chi_{n}^{4}}\right)}^{\frac{1}{7}}\;Y\left(4\right),$$
where Y(4)=0.467138.

The relation between $Re_{*}$ and Re numbers is provided by (Equation 25) formula, which takes the following form for n=4:(35)$$Re=\frac{4Y(n)}{{\left(5\chi_{n}^{4}\right)}^{\frac{1}{7}}}{\left(Re_{*}\right)}^{\frac{8}{7}}=13.985077\;{\left(Re_{*}\right)}^{\frac{8}{7}},$$
and backwards:(36)$$Re_{*}=\frac{5^{\frac{1}{7}}{\left(\chi_{n}\right)}^{\frac{4}{7}}}{4Y(4)}Re^{\frac{7}{8}}=0.102335Re^{\frac{7}{8}}.$$

For the average velocity over the cross section of the pipe and according to (Equation 28), there is:$$V_{av}^{2}=25.2844\;Re^{\frac{1}{4}},\;\;\;V_{av}=5.0283596\;Re^{\frac{1}{8}}.$$

The resistance coefficient, according to (Equation 28) for n=4 and $\chi_{n}=0.019746$ will be as follows:$$\lambda =2^{\frac{3}{2}}\frac{5^{\frac{1}{4}}}{0.467138^{\frac{7}{4}}}\frac{0.019746}{Re^{\frac{1}{4}}}.$$

Thus, we obtained the Blasius’ Formula (Equation 2), $\lambda =\frac{0.3164}{Re^{1/4}}$.

Rheological relation (Equation 5) for n=4 and $\chi_{n}=0.019746$ allows the description of the turbulent flow in the “Blasius” range of Reynolds numbers, which is $10^{4}<Re<10^{6}$ for a flow in a pipe. At the same time, unlike the modern approaches to turbulence modelling, the velocity profile here is obtained in the form of a single curve similar to the Poiseuille profile in the laminar flow regime. In case we need to describe flow outside of the “Blasius” Reynolds numbers range, it is possible to use, for instance, the power value n=6 and $\chi_{n}=0.00910904$. As a result, it is possible to obtain the Prandtl–Nikuradze resistance curve as the envelope of family of resistance curves for various values of the parameters *n* and $\chi_{n}$, consisting, for simplicity, of a set of piecewise-smooth functions. For smooth conjugation of solutions, it is necessary to use additional conditions for their connection.

In dimension form, the velocity profile after some transformations of the expression (Equation 33), taking into account the expression for the friction velocity and the fact that $\eta =1-\frac{r}{R}$, takes the form:(37)$$u=0.93677\frac{1}{\nu^{\frac{1}{7}}}{\left(\frac{1}{\rho }\frac{\Delta p}{l}\right)}^{\frac{4}{7}}{\left(R^{5}-r^{5}\right)}^{\frac{1}{7}}.$$

This formula can be obtained in a common way, operating with dimensional quantities. Considering that the shear stress is τ<0, $\frac{dp}{dz}<0$, $\left|\frac{dp}{dz}\right|=\frac{\Delta p}{l}$ for a flow in the pipe, from the comparison of expressions for τ from the rheological relation (Equation 32) and from the equation of fluid motion in stresses, we can write down that:$$\chi_{n}\nu^{\frac{1}{4}}{\left(\frac{du^{7}}{dy}\right)}^{\frac{1}{4}}=\frac{1}{2}\left|\frac{dp}{dz}\right|\left(R-y\right),$$
hence the differential equation follows:$$\frac{du^{7}}{dy}=A{\left(R-y\right)}^{4},$$
where for brevity it is denoted that:(38)$$A=\frac{1}{\nu {\left(2\chi_{n}\rho \right)}^{4}}{\left(\frac{\Delta p}{l}\right)}^{4}.$$

Integration of this differential equation considering the no-slip condition provides:$$u^{7}=\frac{A}{5}\left(R^{5}-{\left(R-y\right)}^{5}\right),$$
or, taking into account that R−y=r, where *r* is the radial coordinate measured from the pipe axis:$$u^{7}=\frac{A}{5}\left(R^{5}-r^{5}\right).$$

Hence the velocity profile is as follows:$$u={\left(\frac{A}{5}\right)}^{\frac{1}{7}}{\left(R^{5}-r^{5}\right)}^{\frac{1}{7}}.$$

After substituting the “*A*” parameter into that expression, according to (Equation 38), we have:(39)$$u={\left(\frac{1}{80\chi_{n}^{4}}\right)}^{\frac{1}{7}}\frac{1}{\nu^{\frac{1}{7}}}{\left|\frac{\frac{1}{\rho }dp}{dz}\right|}^{\frac{4}{7}}{\left(R^{5}-r^{5}\right)}^{\frac{1}{7}},$$
which is equivalent to the formula (Equation 37).

All power-law rheological relations, including V. V. Novozhilov’s theory [19], lead to the derivative of the velocity on the wall being an infinitely large value:$$\frac{du}{dy}{|}_{y=0}=\infty ,$$
but at the same time $\frac{du^{7}}{dy}{|}_{y=0}=AR^{4}$, i.e., this value is finite, which provides a finite value of the velocity *u* on the wall. The non-dimensional velocity profile in the pipe according to the expression below:(40)$$\frac{u}{u_{max}}=\frac{V}{V_{max}}={\left(1-{\left(1-\eta \right)}^{5}\right)}^{\frac{1}{7}},$$
is presented in Figure 1, where the dots specify the experimental data observed by L.S. Artjushkov [11] for the Reynolds number $Re=3.24\cdot 10^{6}$.

There is no dependency on the Reynolds number for this profile, as well as for the laminar profile as the Blasius laws for the resistance coefficient and the corresponding velocity profile are suitable for a limited range of Reynolds numbers ($10^{4}<Re<10^{6}$).

## 5. Flows at Other Values of the *n* Power

Power-law rheological relation (Equation 5) can be used to describe flows of weakly concentrated aqueous solutions of polymers demonstrating the Toms effect [20]. This effect is connected with the deviation of the resistance curve from the Prandtl–Nikuradze resistance law with access to a section equidistant to the resistance curve $\lambda =\frac{64}{Re}$ for the laminar flow regime. The range of Reynolds numbers where the Toms effect occurred depends on the type of polymer, its concentration, and the diameter of the pipe. Numerical values for this range, depending on these factors, are contained in the article [16]. For the Virk limit curve [21], at which the resistance curves of various polymer concentrations depart after deviating from the turbulent resistance law for smooth pipes, we shall assume n=2 and $\chi_{n}=0.032146$, which provides:(41)$$\tau =\rho \chi_{n}{\left(\nu \frac{du^{3}}{dy}\right)}^{\frac{1}{2}}.$$

Hence, the velocity profile in non-dimensional form, according to (Equation 22) is as follows:(42)$$V={\left(\frac{Re_{*}}{3\chi^{3}}\right)}^{\frac{1}{3}}{\left(1-{\left(1-\eta \right)}^{3}\right)}^{\frac{1}{3}}.$$

Then, it is possible to determine expressions for $V_{av}$, Re, $Re_{*}$ and the resistance law can be obtained in the form below:(43)$$\lambda =\frac{0.87}{Re^{\frac{1}{2}}}.$$

This law is limited to flows of weakly concentrated aqueous solutions of polymers demonstrating the Toms effect [22], and the corresponding resistance curve is called the Virk limit curve. Note that in the article [21], this curve is described by the $\lambda =\frac{2.36}{Re^{0.58}}$ equation, with which the relation (Equation 43) practically coincides in the range of Reynolds numbers, where the Toms effect proves itself.

For a flow in rough pipes, one can take n=1000, which gives an almost “horizontal line” in coordinates “logarithm of Reynolds number – logarithm of resistance coefficient”, i.e., “$log_{10}Re-log_{10}\left(100\lambda \right)$”. The power relation (Equation 5) when n=1000 provides for shear stress the expression below:$$\tau =\rho \chi_{n}{\left(\nu \frac{du^{1999}}{dy}\right)}^{\frac{1}{1000}}.$$

Here, for the best agreement with experimental data, one should use $\chi_{n}=\frac{0.02}{m^{1/3}}$, where $m=\frac{R}{k}$ is the roughness parameter, *k* is the bumps of roughness height. Hence, the velocity profile in non-dimensional coordinates, according to the expression (Equation 22), is:$$V={\left(\frac{Re_{*}}{1001\chi^{1000}}\right)}^{\frac{1}{1999}}{\left(1-{\left(1-\eta \right)}^{1001}\right)}^{\frac{1}{1999}}\approx \frac{1}{\sqrt{\chi_{n}}}{\left(1-{\left(1-\eta \right)}^{1001}\right)}^{\frac{1}{1999}}.$$

The average value of the non-dimensional velocity:$$V_{av}=\frac{2}{\sqrt{\chi_{n}}\;Y\left(1000\right)}=\frac{2}{\sqrt{\chi_{n}}}\cdot 0.5=\frac{1}{\sqrt{\chi_{n}}}.$$

For the roughness parameter values m=15;60;507, $V_{av}=11.6;14.4;20.2$ could be computed, which is in good agreement with the experimental data [23]. The relation between Reynolds numbers for average and friction velocities for flow in rough pipes is as follows:$$Re=2Re_{*}V_{av}=\frac{2Re_{*}}{\sqrt{\chi_{n}}},\;\;\;Re_{*}=\frac{\sqrt{\chi_{n}}}{2}Re.$$

According to expression (Equation 28), the resistance coefficient is:$$\lambda =\frac{8\chi_{n}}{Re^{0.001}},$$
or approximately $\lambda ≅\frac{0.16}{m^{1/3}}$, which is in satisfactory agreement with experimental data for flow in rough pipes [1,23].

The power-law rheological relation (Equation 5) for other values of the power *n* can be used to describe the behavior of non-Newtonian fluids by calculating the $\chi_{n}$ based on the results of experiments.

## 6. Turbulent Flow in a Flat Channel

Let us consider the solution to the problem of turbulent fluid flow in a flat channel (slot) bounded by two parallel planes, with a distance between them of 2 h. Transversal coordinate *y* will be measured from the lower wall 0≤y≤2 h. For n=4 and $\chi_{n}=0.019746$ the equation of motion (Equation 15) has the form:(44)$$-\frac{dp}{dx}+\rho \chi_{n}\nu^{\frac{1}{4}}\frac{d}{dy}{\left(\left|\frac{du^{7}}{dy}\right|\right)}^{\frac{1}{4}}=0.$$

As the derivative of the velocity changes sign on the channel axis for such flow, then the velocity modulus will be equal to $\frac{du}{dy}$ when 0≤y≤h and $-\frac{du}{dy}$ when h≤y≤2 h.

For the lower flow zone, the equation of motion has the form below:(45)$$-\frac{dp}{dx}+\rho \chi_{n}\nu^{\frac{1}{4}}\frac{d}{dy}{\left(\frac{du^{7}}{dy}\right)}^{\frac{1}{4}}=0.$$

The first integration gives the next expression:$${\left(\frac{du^{7}}{dy}\right)}^{\frac{1}{4}}=\frac{1}{\rho \chi_{n}\nu^{\frac{1}{4}}}\frac{dp}{dx}y+C_{1}.$$

From the condition of symmetry of the flow relative to the channel axis, the below condition is satisfied:$$y=h,\;\;\;\frac{du^{7}}{dy}=0,$$
then $C_{1}=-\frac{1}{\rho \chi_{n}\nu^{0.25}}\frac{dp}{dx}h$, and as a result we have:$$\frac{du^{7}}{dy}={-\left(\frac{dp}{dx}\right)}^{4}\frac{1}{\rho \chi_{n}\nu^{\frac{1}{4}}}{\left(h-y\right)}^{4}.$$

The second integration leads to the next expression:$$u^{7}=-{\left(-\frac{dp}{dx}\right)}^{4}\frac{1}{{\left(\rho \chi_{n}\right)}^{4}\nu }\;\;\frac{1}{5}{\left(h-y\right)}^{5}+C_{2}.$$

The no-slip condition provides u=0 for y=0, therefore the arbitrary constant $C_{2}$ is:$$C_{2}={\left(-\frac{dp}{dx}\right)}^{4}\frac{h^{5}}{5{\left(\rho \chi_{n}\right)}^{4}\nu }.$$

Hence:$$u^{7}=\left[{\left(\frac{1}{\rho \chi_{n}}*\frac{dp}{dx}\right)}^{4}\frac{1}{5\nu }\right]\left[h^{5}-{\left(h-y\right)}^{5}\right],$$
and the velocity profile is determined by the expression:(46)$$u={\left[{\left(\frac{1}{\rho \chi_{n}}\frac{dp}{dx}\right)}^{4}\frac{1}{5\nu }\right]}^{\frac{1}{7}}{\left[h^{5}-{\left(h-y\right)}^{5}\right]}^{\frac{1}{7}}.$$

For the upper zone, when h≤y≤2 h, i.e., 1≤η≤2, the motion Equation (Equation 12), taking into account that the modulus of the derivative will be equal to $-\frac{du}{dy}$, leads to the velocity profile in the following form:$$u={\left[{\left(\frac{1}{\rho \chi_{n}}\frac{dp}{dx}\right)}^{4}\frac{1}{5\nu }\right]}^{\frac{1}{7}}{\left[h^{5}-{\left(y-h\right)}^{5}\right]}^{\frac{1}{7}},$$
which is symmetrical to the lower profile relative to the channel axis. If the non-dimensional quantities are introduced, as non-dimensional coordinate η and non-dimensional velocity *V*, in terms of the friction velocity $V_{*}=\sqrt{\frac{\left|\tau_{w}\right|}{\rho }}=\sqrt{\frac{1}{p}\left(-\frac{dp}{dx}\right)h}$, i.e.,
$$\eta =\frac{y}{h},\;\;\;0\leq \eta \leq 2,\;\;\;V=\frac{u}{V_{*}},$$
and also $Re_{*}=\frac{hV_{*}}{\nu }$, which is the friction velocity Reynolds number, then the expression for the non-dimensional velocity profile can be obtained for 0≤η≤1:(47)$$V={\left(\frac{Re_{*}}{5{\chi_{n}}^{4}}\right)}^{\frac{1}{7}}{\left[1-{\left(1-\eta \right)}^{5},\right]}^{\frac{1}{7}},$$
coinciding with the expression for the velocity profile in a circular pipe. The velocity profile for 1≤η≤2 is as follows:$$V={\left(\frac{Re_{*}}{5{\chi_{n}}^{4}}\right)}^{\frac{1}{7}}{\left[1-{\left(\eta -1\right)}^{5}\right]}^{\frac{1}{7}}.$$

The average value of the velocity over the channel cross section is determined by the expression:$$V_{av}=2\int_{0}^{1}V\left(\eta \right)\eta d\eta ,$$
which, after substituting the profile (Equation 47) into it, provides:(48)$$V_{av}=2{\left(\frac{Re_{*}}{5{\chi_{n}}^{4}}\right)}^{\frac{1}{7}}Y,$$
where *Y* is:$$Y=\int_{0}^{1}{\left[1-{\left(1-\eta \right)}^{5}\right]}^{\frac{1}{7}}\eta d\eta .$$

It is expressed in terms of hypergeometric functions and equals Y=0.495849.

The resistance coefficient is determined by the formula:$$\lambda =\frac{8\left|\tau_{w}\right|}{\rho u_{av}^{2}}=\frac{8}{V_{av}^{2}},$$
where the value $Re_{*}$ can be expressed in terms of the Reynolds number $Re=\frac{2hV_{av}}{\nu }$:$$Re=2Re_{*}V_{av}=\frac{4Y}{{\left(5{\chi_{n}}^{4}\right)}^{\frac{1}{7}}}Re^{\frac{8}{7}},\;\;\;Re_{*}=\left(\frac{5^{\frac{1}{7}}{\chi_{n}}^{\frac{4}{7}}}{4Y}\right)Re^{\frac{7}{8}}.$$

Since n=4 and $\chi_{n}=0.019746$ for this turbulent flow, then considering the Y=0.495849 there are:$$Re_{*}=0.0943795Re^{\frac{7}{8}},\;\;\;V_{av}=5.297762Re^{\frac{1}{8}}.$$

Then, finally, for a turbulent flow in a flat channel, the resistance coefficient will be as follows:(49)$$\lambda =\frac{0.285}{Re^{\frac{1}{4}}},$$
which is in good agreement with experimental data [8,19]. Let us note that the expression (Equation 49) can also be obtained from the formula (Equation 28).

## 7. Plane Couette Flow

For the unpressurized plane Couette flow, when the lower plate is stationary and the upper one moves parallel to it with the $V_{0}$ velocity, the motion Equation (Equation 15) for a given flow leads to the following common differential equation:$$\rho \chi_{n}{\left(\nu \frac{du^{7}}{dy}\right)}^{\frac{1}{4}}=C,$$
where the constant of integration *C* is the shear stress. After introducing non-dimensional parameters:$$\eta =\frac{y}{h};\;\;0\leq \eta \leq 1,\;\;\;V=\frac{u}{V_{0}},\;\;\;Re=\frac{V_{0}h}{v},$$
where *h* is the distance between the plates, this equation takes the form below:(50)$$\frac{dV^{7}}{d\eta }={\left(\frac{C}{\rho V_{0}^{2}\chi_{n}}\right)}^{4}Re.$$

The boundary conditions will be:η=0:V=0;η=1:V=1.

For the Couette flow, when the lower wall is stationary and the upper one moves with the velocity *V* the velocity profile has an inflection point at which the second derivative changes sign. Therefore, it is convenient to divide the flow region into two zones: close to the lower wall (with velocity $V_{lower}$) and near the upper wall (with velocity $V_{upper}$), and the obtained solutions for these zones must be joined. At the joint point for the velocity profiles near the lower and upper walls, the conditions of equality of the velocities and their first derivatives are satisfied:(51)$$\eta =\eta_{w},\;\;\;V_{lower}=V_{upper},\;\;\;{\left.\frac{dV}{d\eta }\right|}_{lower}={\left.\frac{dV}{d\eta }\right|}_{upper}.$$

Equation (Equation 50) describes the flow near the stationary lower wall, the boundary condition for which will be the no-slip condition:(52)η=0,V=0.

For brevity, the right-hand side of Equation (Equation 50) is denoted as *a*:(53)$$a={\left(\frac{C}{\rho V_{0}^{2}\chi_{n}}\right)}^{4}\;Re.$$

Then Equation (Equation 50) can be written as:(54)$$\frac{dV^{7}}{d\eta }=a.$$

The parameter *a* is still unknown and should be found in the process of solving the problem. Integration of this equation considering the boundary condition (Equation 52) gives an expression for the velocity profile near the lower wall:(55)$$V={\left(a\eta \right)}^{\frac{1}{7}}.$$

Since the proposed generalization of Newton’s formula is not invariant under the Galilean transformation, the velocity profile near the upper wall can be found by assuming it to be stationary, and the lower one moving relative to it with the same velocity, but in the opposite direction. The transversal coordinate *s* will now be measured from the upper wall towards the lower one. Then, for the non-dimensional fluid velocity *w* relative to the upper plate, an equation similar to (Equation 55) can be written, taking into account the lower plate moving in the opposite direction:(56)$$w=-{\left(as\right)}^{\frac{1}{7}}.$$

In terms of *V* and η, when the origin is related to the lower wall, taking into account that
(57)s=1−η,w=V−1,
the expression (Equation 56) takes the following form:$$V-1=-{\left(a\left(1-\eta \right)\right)}^{\frac{1}{7}}.$$

From which the fluid velocity near the upper wall is:(58)$$V=1-{\left(a\left(1-\eta \right)\right)}^{\frac{1}{7}}.$$

Solutions for the upper and lower walls velocities are to be joined. At the joint point $\eta_{c}$ there is the equality of velocities (Equation 55) and (Equation 58) and their derivatives:(59)$${\left(a\eta_{c}\right)}^{\frac{1}{7}}=1-\;{\left(a\left(1-\eta_{c}\right)\right)}^{\frac{1}{7}},$$
(60)$$\frac{a^{\frac{1}{7}}}{{\eta_{c}}^{\frac{6}{7}}}=\frac{a^{\frac{1}{7}}}{{\left(1-\eta_{c}\right)}^{\frac{6}{7}}}.$$

From relation (Equation 60), the coordinate of the joint point is $\eta_{c}=0.5$, which was expected because of the problem symmetry. From Equation (Equation 59), $a=1/2^{6}$ is obtained.

The velocity profile will have an inflection point η=0.5, at which the second derivative changes its sign.

Thus, the velocity profile for the considered Couette flow has the form:(61)$$V=\;\;\left\{\begin{array}{c} {\left(a\eta \right)}^{\frac{1}{7}},\;\;\;if\;\;\;0\leq \eta \leq 0.5\\ 1-{\left[a\left(1-\eta \right)\right]}^{\frac{1}{7}},\;\;\;if\;\;\;0.5\leq \eta \leq 1, \end{array}\right.$$
or, considering that $a=1/2^{6}$, the velocity profile can be expressed as:(62)$$V=\;\;\left\{\begin{array}{c} 0.552\;\eta^{\frac{1}{7}},\;\;\;if\;\;\;0\leq \eta \leq 0.5\\ 1-0.552\;\eta^{\frac{1}{7}},\;\;\;if\;\;\;0.5\leq \eta \leq 1. \end{array}\right.$$

It coincides with that given in the monograph [6] and is in good agreement with the experimental data from this monograph. The shear stress *C* according to the notation (Equation 53) can be expressed in terms of the Reynolds number, after which the resistance coefficient takes the form:(63)$$\lambda =\frac{2C}{\rho V_{0}^{2}}=\frac{2\chi_{n}a^{\frac{1}{4}}}{Re^{\frac{1}{4}}}=\frac{0.014}{Re^{\frac{1}{4}}},$$
which is in the satisfactory agreement with the experimental data by [6].

Thus, the proposed set of equations of turbulent fluid motion can be useful for at least obtaining the preliminary and estimated characteristics of turbulent flow before starting numerical simulations using differential turbulence models. The proposed rheological ratio for some power values can also be used to describe the behavior of power-law fluids, as well as fluids with small additives of polymers when the Toms effect is observed.

## 8. Conclusions

Thus, depending on the power value in the proposed rheological ratio, one can come either to a description of a laminar flow regime (in the form of the Navier–Stokes equation), or to a description of a flow in a turbulent regime. In the latter case, there is a system of differential equations with the no-slip boundary condition, which significantly differs from that for the laminar flow regime, but which also allows the finding of the analytical solutions for simple shear flows and obtain the Blasius resistance law for a flow in a pipe. Therefore, the considered approach to a solution to turbulent flows problems compares favorably with modern differential turbulence models. The solutions obtained for the problems of turbulent fluid flows in terms of the velocity profiles and resistance coefficients are in satisfactory agreement with experimental data. However, this agreement is slightly worse than when differential turbulence models are used, which is connected with more accurate results due to many empirical constants in such models, often selected for solving a specific problem. It also necessary to note that the proposed generalization of Newton’s formula, as all other power rheological relations, is not invariant with respect to the Galilean transformation, and therefore the area of possible applications of this generalization is limited to cases of flow over stationary walls. Thus, the proposed set of equations of turbulent fluid motion can be useful at least for obtaining the preliminary and estimated parameters of turbulent flow over stationary walls before numerical simulations using differential turbulence models. The proposed rheological ratio for some power values can also be used to describe the behavior of power-law fluids, as well as fluids with small additives of polymers where the Toms effect is observed.

## Figures and Tables

**Figure 1 polymers-14-03308-f001:**
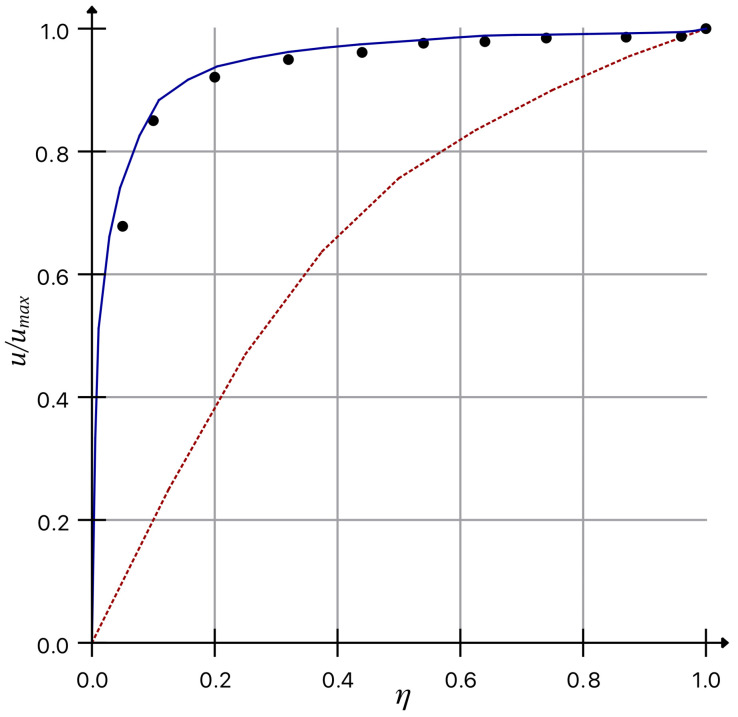
Non-dimensional velocity profile of a turbulent flow in comparison with laminar Poiseuille profile (dash line) and experimental results by L.S. Artjushkov [11] (shown as dots).

**Table 1 polymers-14-03308-t001:** Values of $Y\left(n\right)$ function.

*n*	1	2	3	4	5	20
Y(n)	0.25	0.403067	0.447761	0.467138	0.477358	0.498156

## Data Availability

Not applicable.

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
