# Peer review of "Analytical Solutions for Simple Turbulent Shear Flows on a Basis of a Generalized Newton’s Law"

_polymers, 2022, doi:10.3390/polym14163308_

Round 1

Author Response

Dear collegue,

Regarding to your review.

  1. In the future, we plan to solve non-stationary problems for turbulent flows, for inctance, based on the method of immersed boundaries to ensure the fulfillment of boundary conditions on the surface of streamlined bodies in the obtained numerical solution. With help of this approach, the problem of a turbulent flow around a three-dimensional cylinder was numerically solved (Abalakin I.V., Duben A.P., Zhdanova N.S., Kozubskaya T.K. MODELING OF UNSTATIONARY TURBULENT FLOW AROUND A CYLINDER BY THE METHOD OF IMMEDED BOUNDARY. MATHEMATICAL MODELING 2018, vol. 30, no. 5, pp. 117-133.). The numerical solution obtained was compared with large amount of experimental and calculated data. It is also possible to solve non-stationary problems using grids consistent with the boundary.
  2. No-slip conditions allow to obtain results, which are in good agreement with experimental data. It is also possible to use alternative conditions, but it is required additional researches.
  3. The introduction has been updated. 
  4. English language was revised

Thank you very much for the positive review!

Reviewer 2 Report

The manuscript entitled Analytical Solutions for Simple Turbulent Shear Flows on a basis of a generalized Newton's law is reviewed. I commented as follows;

1.Captions in table and figure is poor. The author should revise them.

2.Because only theoretical model was expressed, validity was not obtained. The author should revise it.

3.The developed model should be applied to the experimental results.

4.The many symbols and letters are used. The author should summarize them as a nomenclature.

Author Response

Dear Colleague,

Thank you for your review.

Regarding to your comments:

  1. The captions were revised and updated
  2. The reliability of the calculation results is confirmed by the obtained formulas for the drag coefficients, which were compared with formulas generally recognized in modern literature and are in good agreement 
  3. The presented research shows that all flow regimes and all kinds of rheology can be approximately described by the only rheological relation, which is a power-law generalization of Newton's formula for the flow of a viscous fluid, by changing values of the power $n$ and parameter $\chi_n$ in this relation accordingly. Simulation of a flow in a pipe showed that the obtained computational results are in good agreement with experimental data by L. S. Artjushkov. Possible use of partial differential equations of the same type as equation (12) is related to the study of
    • flows of non-Newtonian fluids, including those that, when small additives of polymers are introduced into the flow, exhibit the Toms effect;
    • transport of dissolved substances with longitudinal and transverse diffusion in time and space from an impulse source;
    • diffusion flow of radon through the concrete foundation of buildings, etc.
  4.  All symbols and letters are described in the text and the same letters are used in different sections of the manuscript. Perhaps it is not necessary to introduce separate nomenclatures section, what is your opinion? 
  5. English language was revised, introduction was updated

Round 2

Reviewer 2 Report

No comment.